# Sinonasal Squamous Cell Carcinoma, a Narrative Reappraisal of the Current Evidence

**DOI:** 10.3390/cancers13112835

**Published:** 2021-06-07

**Authors:** Marco Ferrari, Stefano Taboni, Andrea Luigi Camillo Carobbio, Enzo Emanuelli, Roberto Maroldi, Paolo Bossi, Piero Nicolai

**Affiliations:** 1Section of Otorhinolaryngology—Head and Neck Surgery, Department of Neurosciences, University of Padua—“Azienda Ospedaliera di Padova”, 35128 Padua, Italy; stefano.taboni@unipd.it (S.T.); andrealc.carobbio@gmail.com (A.L.C.C.); piero.nicolai@unipd.it (P.N.); 2Guided Therapeutics (GTx) Program International Scholarship, University Health Network (UHN), Toronto, ON M5G 2C1, Canada; 3Technology for Health (PhD Program), Department of Information Engineering, University of Brescia, 25123 Brescia, Italy; 4Artificial Intelligence in Medicine and Innovation in Clinical Research and Methodology (PhD Program), Department of Clinical and Experimental Sciences, University of Brescia, 25123 Brescia, Italy; 5Section of Otorhinolaryngology—Head and Neck Surgery, Department of Neurosciences, University of Padua—“Ospedale di Treviso”, 31100 Treviso, Italy; enzo.emanuelli@aulss2.veneto.it; 6Unit of Radiology, Department of Medical and Surgical Specialties, Radiologic Sciences, and Public Health, University of Brescia—“ASST Spedali Civili di Brescia”, 25123 Brescia, Italy; roberto.maroldi@unibs.it; 7Unit of Medical Oncology, Department of Medical and Surgical Specialties, Radiologic Sciences, and Public Health, University of Brescia—“ASST Spedali Civili di Brescia”, 25123 Brescia, Italy; paolo.bossi@unibs.it

**Keywords:** sinonasal cancer, nasal cavity, paranasal sinuses, squamous cell carcinoma, diagnosis, staging, treatment, surgery, chemotherapy, radiotherapy

## Abstract

**Simple Summary:**

Sinonasal squamous cell carcinomas are a group of diverse tumors affecting the nasal cavity and paranasal sinuses. As a direct consequence of their rarity and heterogeneity, diagnosis is challenging, and treatment does not follow universally accepted protocols. Though surgery represents the mainstay of treatment, neoadjuvant and adjuvant therapies have pivotal roles in improving outcomes of patients treated with curative intent. Indications to endoscopic surgery have been expanding over the last three decades, but a considerable number of patients affected by sinonasal squamous cell carcinoma still need open surgical procedures. Management of the neck in patients affected by sinonasal squamous cell carcinoma is controversial. Curative-intended treatment of recurrent and/or metastatic tumors, alongside palliation of uncurable cases, represent poorly explored aspects of this disease.

**Abstract:**

Sinonasal squamous cell carcinoma is a rare tumor affecting the nasal cavity and paranasal sinuses. Several aspects of this disease, ranging from epidemiology to biology, pathology, diagnosis, staging, treatment, and post-treatment surveillance are controversial, and consensus on how to manage this sinonasal cancer is lacking. A narrative literature review was performed to summarize the current evidence and provide the reader with available data supporting the decision-making process in patients affected by sinonasal squamous cell carcinoma, alongside the authors’ personal opinion on the unsolved issues of this tumor. The review has highlighted several advances in molecular definition of epithelial cancers of the sinonasal tract. Surgery represents the pivot of treatment and is performed through an endoscopic transnasal approach whenever feasible. Open surgery is required for a large proportion of cases. Reconstruction of the defect follows principles of skull base and cranio-maxillo-facial reconstruction. Chemotherapy is given as neoadjuvant treatment or concomitantly to radiotherapy. Photon-based radiation therapy has a crucial role in the adjuvant setting. Particle therapy is providing promising results. Management of the neck should be planned based on the presence of clinically appreciable metastases, primary tumor extension, and need for recipient vessels. Biotherapy and immunotherapy are still underexplored therapeutical modalities.

## 1. Introduction and Epidemiology

Squamous cell carcinoma (SCC) of the sinonasal tract (SNSCC) is “a malignant epithelial neoplasm arising from the surface epithelium lining the nasal cavity and paranasal sinuses and exhibiting squamous differentiation” [1]. Despite this relatively simple definition, SNSCC includes a wide group of tumors with heterogeneous biological features, as witnessed by the fact that its genetics showed partial overlap with respect to other sinonasal cancers, such as sinonasal undifferentiated carcinoma (SNUC) and neuroendocrine carcinomas (NECs) [2]. As opposed to other sites of the head and neck (HN), SNSCC does not rarely display features pertaining to other rare sinonasal tumors, thus mirroring the unparalleled biological heterogeneity that is observed in cancers of the sinonasal tract within the context of a single histology. Differently to other HNSCCs, the putative etiological role of tobacco smoking is based upon frail and dated evidence [3], and alcohol consumption has not been demonstrated as being a risk factor for SNSCC. Several patients affected by SCC of the nasal vestibule had a history of tobacco smoking [4,5], which probably led some authors to find an association with this risk factor when SCCs of the nasal vestibule, nasal cavity, and paranasal sinuses are grouped altogether. On the other hand, occupational exposure to leather dust, chromium, arsenic, asbestos, and/or welding fumes was found to increase the risk of developing an SNSCC [6,7,8]. Similar to some HNSCCs, transcriptionally active human papillomavirus (HPV) has been found in some SNSCCs, though the causative role of viral infection and its prognostic implications are still a matter of debate [9]. However, recent publications converge in stating that HPV probably has a role in determining SNSCC and implies a better prognosis compared to HPV-negative tumors [10,11,12]. In a similar fashion, Epstein–Barr virus (EBV) has been detected in a relevant proportion of SNSCCs [13,14]. However, since a similar proportion of EBV infection was observed in nasal polyps, its role in SNSCC carcinogenesis is questionable [14].

A recent study on 4994 SNSCCs registered at the United States National Cancer Institute’s Surveillance, Epidemiology, and End Results revealed an incidence of 0.32 new cases/100,000 habitants/year, with a steadily declining trend of −2.6%/year over the last 3 decades [15]. The male-to-female ratio, in terms of incidence, was 2.3:1, and almost 80% of patients were 55 years old or older. In patients with neither regional nor distant metastasis, 5-, 10-, 15-, and 20-year overall survival (OS) rates were 82.9%, 73.8%, 60.6%, and 43.7%, respectively. SNSCCs with nodal metastases and those with distant disease showed 5-, 10-, 15-, and 20-year OS rates of 41.1%, 32.8%, 26.2%, and 22.5%, and 29.2%, 19.8%, 18.3%, and 17.1%, respectively.

Most relevant up-to-date evidence on pathological, diagnostic, and therapeutical characteristics of SNSCC are hereby summarized in a narrative fashion. For those aspects lacking in evidence and consensus, the authors will report their experience-based opinion, yet recognizing it represents the lowest level of evidence in the scientific literature.

## 2. Pathological Features

SCC is the most common malignant neoplasm of the nasal cavity and paranasal sinuses. Approximately 50–60% of all sinonasal cancers are SCC [16]. SNSCC is classified as either keratinizing (KSCC) or non-keratinizing (NKSCC): approximately 50% are KSCCs and 30% are NKSCC, the remaining 20% being represented by other variants of SCC [17]. KSCC exhibits irregular nests and cords of eosinophilic cells demonstrating keratinization and inducing a desmoplastic stromal reaction, similarly to conventional SCC of other HN sites, and its grade is classified as well, moderately or poorly differentiated [1]. NKSCC is characterized by a distinctive ribbon-like growth pattern, with absent to limited maturation. Historically, NKSCC was also called “Schneiderian carcinoma”, “transitional cell carcinoma”, or “cylindrical cell carcinoma” in relation to the patterns of growth and cell morphology [17,18]. Currently, these terms are considered as obsolete synonyms of NKSCC [1]. There is no established role for tumor grading in NKSCC, and it is unclear whether the prognosis differs from that of KSCC [1]. Other morphological variants of SNSCC, each with special histopathological appearance and features, account for approximately 20% of cases. The five main variants of SCC are the following: adenosquamous, spindle cell (also referred to as “sarcomatoid”), basaloid, papillary, and verrucous SCC. The aforementioned variants are associated with a 5-year disease-specific survival (DSS) of 15%, 32%, 56%, 62%, and 70%, respectively [19]. The 5-year DSS of patients affected by conventional SNSCC (i.e., considering both KSCC and NKSCC) is 45% [16].

The maxillary sinus is the most common site affected by SCC of the sinonasal tract (approximately 60% of the cases), followed by the nasal cavity (25%), and ethmoidal complex (15%) [16]. SCCs arising from the maxillary sinus present a different behavior compared to those originating from the nasoethmoidal complex. For instance, though regional lymph node metastases are overall uncommon in SNSCC, maxillary SCCs classically present a higher rate of nodal involvement compared to nasoethmoidal SCCs, probably due to the different lymphatic drainage of the two primary sites [20].

Malignant transformation of Schneiderian papillomas is a known phenomenon, particularly for an inverted papilloma, whose transformation rate to SCC is reported as being as high as 10% [21]. Approximately 15% of all SNSCCs are either synchronously or metachronously associated with papilloma [22,23]. Tobacco smoking is thought to be a factor promoting transformation of Schneiderian papillomas into SNSCC [24].

Interestingly, approximately 20–25% of SNSCCs test positive for HPV. The majority of these are in the NKSCC cohort, which is histomorphologically similar to HPV-positive SCC of the oropharynx. The implications of this association are not clear yet. However, recent studies suggested a more favorable outcome in HPV-positive SNSCCs [11,25,26,27]. HPV-related multiphenotypic sinonasal carcinoma (originally called “HPV-related carcinoma with adenoid cystic-like features”) is a recently recognized variant of sinonasal carcinoma showing histologic features of surface dysplasia and an adenoid cystic carcinoma-like pattern, and presenting a strong association with HPV, particularly HPV-33 [28]. Even though these tumors are often large and destructive, they tend to exhibit a relatively indolent behavior. In fact, although local recurrence is frequent, distant metastasis and death related to the disease are very uncommon [28,29].

Another pathological entity to be considered is NUT carcinoma [1,30]. Histologically, NUT carcinoma grows as nests and sheets of undifferentiated cells within the sinonasal submucosa. NUT carcinoma is highly infiltrative, does not display any carcinoma in situ component, and shows necrosis, high mitotic rate, and an intratumoral acute inflammatory infiltrate [31]. Two histologic hallmarks of NUT carcinoma are monotonous tumor cells and a peculiar pattern of keratinization, often described as “abrupt” [31]. NUT carcinoma is characterized by rearrangements of the NUTM1 gene on chromosome 15q14: the most common fusion partner is BRD4 (in about 70% of cases) [31]. It is considered the most dedifferentiated variant of SNSCC, and is associated with a dismal prognosis and a poor response to any treatment, with a median OS of 6.7 months [32].

The differential diagnosis of poorly differentiated SNSCC is challenging and includes: SNUC (including those characterized by molecular identifiers, such as IDH2-mutant SNUC, SMARCA4-deficient carcinoma, and SMARCB1/INI1-deficient carcinoma), NECs, and adenoid cystic carcinoma [1,33,34,35,36].

## 3. Pre-Treatment Assessment

### 3.1. Diagnosis

Preoperative histologic diagnosis of a given sinonasal lesion is essential to correctly plan a proper treatment strategy. Even to an experienced subspecialty pathologist in referral centers, this could be challenging because of non-specific presentation, overlapping appearance of different neoplasms, and the large miscellany of different possible histologies. Preoperative diagnosis has been reported to be non-concordant to the final histopathological diagnosis in 7–10% of HN cancers [37,38]. Not surprisingly, the highest rate of discrepancy was found in the sinonasal tract, with misdiagnosis observed in 23.8% [38]. In a cohort of 52 sinonasal tumors analyzed by Ganti et al., the rate of discordance was 21.2%, with fibro-osseous lesions and small blue round cell tumors associated with the highest diagnostic discrepancy rate [39]. These findings justify the general recommendation to provide the pathologist with abundant material to formulate the pretreatment diagnosis, as emphasized by Schreiber et al. [40]. Of note, while SCC is the most common histology among cancers arising from the maxillary sinus, a pretreatment diagnosis of SCC has been associated with the lowest diagnostic reliability rate (50%) in tumors centered in the nasoethmoidal area [40].

### 3.2. Staging

Another crucial step in treatment planning for sinonasal tumors is the analysis of local extension, with special reference to skull base, orbit, and infracranial spaces (i.e., pterygopalatine fossa, infratemporal fossa, and parapharyngeal space), which allows a “T category” to be attributed to a given sinonasal cancer according to the latest TNM classification criteria [41]. In this regard, cross-sectional imaging plays an invaluable role, with the gold standard represented by a combination of computed tomography (CT) and magnetic resonance imaging (MRI) (Figure 1 and Figure 2) [42].

Sinonasal tumors can extend to the adjacent regions by transgressing the bony walls of the nasal cavity and paranasal sinuses. Structural changes and reabsorption of these thin bone structures are well depicted by CT by virtue of its high spatial resolution and capability of demonstrating bone density alterations [42,43,44]. On the other hand, gadolinium-enhanced MRI is the dominant tool for structural imaging and is paramount in describing soft tissues involvement, both at the sinonasal region boundaries (periosteum/periorbit and dura mater) [45,46,47] and in the adjacent regions (e.g., brain, orbital content, cavernous sinus, internal carotid artery) [48,49].

SNSCC may occasionally arise from an inverted papilloma, posing an additional issue of differential diagnosis. In this respect, MRI plays a central role in differentiating benign inverted papillomas from SCC; convoluted cerebriform pattern and apparent diffusion coefficient (ADC) value [50], particularly when associated with increased maximum standardized uptake values (SUV) on ^18^F-fluorodeoxyglucose-positron emission tomography (FDG-PET)/CT, are associated with higher probability of malignancy [51,52]. However, this result was not confirmed in other studies [53,54]. ADC value also proved useful in differentiating sinonasal lymphoma and SNSCC [55].

FDG-PET/CT is an essential tool for initial staging of advanced HNSCCs, since it enables simultaneous acquisition of anatomical and metabolic data of the primary tumor, enables an accurate diagnosis of regional neck metastases, and allows the identification of distant metastases in a single session [56]. In the sinonasal tract, different SUV from FDG-PET/CT can play the role of a “metabolic biopsy” in categorizing sinonasal neoplasms into different histopathologic subgroups [57]. FDG-PET/CT has also been suggested for serving as a prognosticator in patients with sinonasal malignancies [58,59]. Absence of pathologic uptake at the first post-treatment PET/CT is highly predictive of significantly better OS, as for other SCCs of the HN [60].

## 4. Treatment

### 4.1. Neoadjuvant Systemic Chemotherapy

The use of neoadjuvant chemotherapy (ChT) (also referred to as “induction ChT”) as part of multimodality treatment for locally advanced cancers of the paranasal sinuses has produced some promising results (Table 1) [61,62].

The main goals of such therapy were reported to be a decreased risk of local recurrence and distant metastases, which are the main patterns of treatment failure of locally advanced sinonasal cancers [71]. In addition, in advanced-stage SNSCC, the use of neoadjuvant ChT has been advocated to improve tumor control and orbital preservation (Figure 3) [72]. Ock et al. showed that patients who achieved a partial response to induction ChT, possibly leading to a T down-classification, had higher chances of orbit preservation and more favorable overall prognosis [63].

In 2011, Hanna et al. reported a series of 46 patients undergoing an induction ChT-including treatment for an SNSCC [62]. ChT regimen was based on a combination of platinum, 5-fluorouracile, ifosfamide, and a taxane. Response to induction ChT (defined by cumulating objective response with stable disease vs. progression of disease) was found to predict a more favorable prognosis than that of patients with disease stability or progression. Hirakawa et al. confirmed that a moderate to marked pathological response to ChT predicts a favorable overall, disease-free, locoregional recurrence-free, and distant recurrence-free survival in patients affected by advanced SNSCC [68]. Abdelmeguid et al. recently reported on the updated series from “The University of Texas MD Anderson Cancer Center”, which accounted for 123 patients [70]. Of note, eight (6.5%) patients also received Cetuximab as part of the neoadjuvant therapy. Non-progression after induction ChT was confirmed as a positive prognostic factor. Definitive locoregional treatment, whether surgery followed by adjuvant (chemo)radiation or (chemo)radiation possibly followed by salvage surgery for residual disease, did not significantly affect OS and DSS. Based on these results, one could conclude that chemoselection of treatment for SNSCC is not as effective as in SNUC [73]. However, in the aforesaid studies, the category of “responders” included a spectrum of variable responses to induction ChT, ranging from complete response to stable disease. In another study assessing the role of chemoselection in SNUCs, Amit et al. [73] classified responders as only patients with an objective response (i.e., either partial or complete response according to Response Evaluation Criteria in Solid Tumors version 1.1), [74] thereby demonstrating the effectiveness of selecting the best locoregional treatment based on response to neoadjuvant ChT. Thus, it can be said that response to neoadjuvant ChT is a reliable prognosticator, whereas more uniform data are required prior to concluding whether or not the best locoregional treatment for advanced SNSCC can be based on chemoselection. Moreover, neoadjuvant ChT was associated with a non-negligible morbidity, with a 34.8% rate of adverse events in a recent study [70]. Therefore, new strategies should be directed towards individualization of the neoadjuvant therapy, on one hand, maximizing the probability to offer systemic treatments to patients who could benefit from them and, on the other, tailoring systemic therapies according to tumor genomic profiles. Several clinical trials are currently investigating the potential benefit conferred by neoadjuvant ChT in sinonasal cancers (e.g., ClinicalTrials.gov (accessed on 1 June 2021) Identifiers: NCT03493425, NCT02099175, NCT02099188, NCT00707473).

### 4.2. Neoadjuvant Intra-Arterial Chemotherapy

Intra-arterial ChT aims at maximizing cytotoxicity while minimizing side effects related to systemic distribution of chemotherapeutic agents. The intra-arterial route permits the delivery of a high concentration of a cytotoxic agent directly to the primary tumor, in an effort to achieve local tumor response, thus omitting mutilating surgery and enabling organ-sparing treatment, while ensuring adequate survival outcomes. The maxillary artery is the preferred vessel to reach the sinonasal tract, thus the intra-arterial infusion finds its natural application for advanced stage SCCs of the maxillary sinus. Nakatani et al. employed neoadjuvant intra-arterial ChT combined with radiotherapy (RT), followed by surgery in 8 patients with advanced maxillary sinus cancer; they reported a response rate of 100% (complete response in 60% of cases), and none of the patients experienced grade 3 to 4 toxicity [75]. Inuyama et al. reported a response rate of 87% (complete response in 47% of cases) in a series of 15 patients treated with neoadjuvant intra-arterial ChT combined with RT, followed by additional treatment according to the extent of residual disease. The reported grade 3 to 4 toxicity was 0% [76]. When considering neoadjuvant intra-arterial ChT in the context of a full treatment, a higher grade 3 to 4 toxicity rate (56% of cases) was reported by Shiga et al. in a series of 25 patients treated with neoadjuvant superselective arterial cisplatin infusion, followed by RT and concomitant superselective arterial cisplatin infusion for maxillary sinus SCC. The same neoadjuvant protocol followed by surgery and adjuvant RT was employed in another 25 patients with no cases of grade 3 to 4 toxicity. The rate of objective response to the entire treatment in the two groups was 92% and 100%, respectively, while a complete response was achieved in 76% and 17% of cases, respectively [77]. The limited number of cases and uncertainty about selection criteria for this procedure prevent any generalization on this therapeutic approach.

### 4.3. Neoadjuvant (Chemo)Radiation

Despite the scarcity of data on this treatment modality, some groups advocated the use of neoadjuvant chemoradiation to optimize treatment morbidity while maintaining adequate oncologic results (Table 1). Radiation dose is most commonly 50 Gy in 25 fractions [66,67,78]. Kreppel et al. did not find a significant difference in OS between 40 Gy and 50 Gy total RT dose [79]. Fu et al. published their findings comparing neoadjuvant and adjuvant RT in non-SCC sinonasal malignancies, reporting a significant improvement in margin status in patients treated with neoadjuvant RT followed by surgery, compared to those treated by upfront surgery and adjuvant RT [78]. Data about the employment of neoadjuvant RT come from a large series of 11,160 patients with sinonasal cancer, analyzed in a national cancer database (NCD) study by Robin et al., who compared different treatments in a subgroup of 3331 patients affected by SNSCC. Compared to surgery alone, neoadjuvant ChT followed by surgery, and neoadjuvant RT followed by surgery, only neoadjuvant chemoradiation was found to increase the rate of clear margins irrespective of tumor site, T, and N category [69]. Moreover, the employment of neoadjuvant therapies conferred an increase in OS in the same large population study. The possible role played by neoadjuvant chemoradiation in organ preservation was investigated by Amsbaugh et al. in a cohort of 20 patients with sinonasal cancer involving the orbit. Among those, 8 were affected by SNSCC. In their series, orbital exenteration was avoided in all patients who received neoadjuvant therapy, whilst all patients who did not receive neoadjuvant therapy underwent an orbital exenteration as part of upfront surgery [67]. Fernstrom et al. published their results concerning the use of neoadjuvant chemoradiation in advanced sinonasal cancer. An orbital sacrifice rate of 7% was reported in their series of 73 patients, of which 42 were SNSCCs [66]. Although one could hypothesize that neoadjuvant (ChT)-RT increases the risk of postoperative complications, sound data confirming this assumption are currently lacking. Mattavelli et al. found that previous ChT and/or RT increases the risk of postoperative cerebrospinal fluid (CSF) leak from 5.1% to 22.2%, though with no statistical significance [80]. As for neoadjuvant ChT, this therapeutical approach deserves to be evaluated in the context of prospective studies with organ preservation approach as an endpoint.

### 4.4. Surgery: Ablation of the Primary Tumor

Surgery has an acknowledged role in the treatment of SNSCC [16,81]. As for all cancers, the principal aim of surgery is complete resection of the tumor. Contrary to other HNSCCs, oncologic surgery of SNSCCs is not supported by a clear cutoff (e.g., 1 cm of unaltered tissue) to be achieved when delineating the tissue to ablate. This owes to both the lack of data and the fact that leaving 1 cm of margin all around tumor boundaries would mean performing a deeply mutilating surgery in the large majority of patients affected by SNSCC. For the latter reason, the concept of “wide margin resection” applied to SNSCC, as well as to sinonasal cancer in general, is rather unrealistic. Moreover, despite an increasing number of patients affected by SNSCC being treated through neoadjuvant ChT-(RT), there are no defined and evidence-based principles to guide the delineation of margins following nonsurgical treatments. However, since several authors advocate for neoadjuvant ChT-(RT) as a strategy to spare the orbit in patients otherwise treated through orbital ablation [49,62,63,64,65], it can be deduced that surgeons do reduce the entity of resection in SNSCCs objectively responding to neoadjuvant treatments.

The current literature is uniform in highlighting the prognostic significance of achieving negative margins, as opposed to leaving microscopic or macroscopic residual disease. Jafari et al. published a large NCD study on 7808 patients affected by SNSCC, demonstrating a stepwise OS decrease ranging from complete resection to microscopically involved margins and macroscopic residual disease [82]. Interestingly, the multivariate, propensity-score-matched analysis performed in the same study showed that ablations ending up with macroscopic residual disease conferred the same chances of survival as nonsurgical treatments. Relying on these data, surgery should be planned in order to avoid macroscopic residual disease.

The surgical approach should be selected based on tumor extension, which dictates the extent of the ablation (Table 2).

The surgical approach should be selected based on tumor extension, which dictates the extent of the ablation. The debate on whether endoscopic or open surgery should be selected has been slowly progressing towards consensus. There is evidence that endoscopic transnasal surgery provides a shorter hospital stay [83] and surgery-to-radiation time [84]. The belief that complications rate was higher for open surgery was instead recently tempered by a meta-analysis [85]. However, endoscopic surgery is generally considered as a less invasive and disfiguring approach, and is usually preferred whenever the completeness of resection is not undermined [83,86,87,88,89]. Indications to purely endoscopic, endoscopic-assisted, and non-endoscopic procedures for sinonasal cancers according to authors’ experience are summarized in Table 2. Of note, some tumor extensions that can be managed with surgery fall under the definition of T4b, which should thus not be automatically considered as an unresectable disease, as also stated in the latest National Comprehensive Cancer Network (NCCN) guidelines (version 3.2021) [90]. The most critical extensions of SNSCC, as for all sinonasal cancers, are those towards the intracranial space [91,92], infracranial spaces [93,94,95,96,97], and orbit [49,65,98,99]. However, advancements in surgical techniques have led to narrow the spectrum of extensions to be considered as genuinely unresectable to a limited number of circumstances (e.g., invasion of both internal carotid arteries, both orbital cavities/optic nerves, or brainstem). More frequently, surgery is excluded for the degree of invasiveness it would imply and the howsoever dismal prognosis. A case-by-case evaluation is thus crucial to correctly identify patients with advanced SNSCC eligible for surgery.

### 4.5. Surgery: Reconstruction after Resection of a Nasoethmoidal/Sphenoidal SCC

In case of an endoscopic resection without transnasal craniectomy, reconstruction is often unneeded. When available, local pedicled mucoperiosteal flaps (e.g., nasoseptal, middle turbinate, and inferior turbinate flaps) [100,101] can be considered to cover large areas of exposed bone, particularly if surgery will be followed by adjuvant treatment. In case of dural exposure and/or resection, skull base reconstruction is mandatory and the technique is dictated by presence and entity of a cerebrospinal fluid (CSF) leak, extension and location of the defect, and availability of flaps [100]. In case of unilateral anterior skull base (ASB) dural resection with preservation of the contralateral septal mucoperiosteum, multilayer graft-based reconstruction (mostly through fascia lata, iliotibial tract (ITT), and/or fat) covered with a septal flip flap represents a safe option [102]. In case of bilateral ASB dural resection and/or unavailability of the septal mucoperiosteum as a donor site, multilayer graft-based reconstruction is effective, with a rate of postoperative CSF leak as low as 5.8% [80,103]. Xeno-/allo-graft dural patch and synthetic dural substitutes are available as alternatives to autologous grafts [104]. In case of very large defects, critical posterior and/or lateral extensions (towards the middle/posterior cranial fossa and the orbital roof, respectively), and/or putatively poor vascularization of defect edges (e.g., previous RT, planned reirradiation, severe infection/necrosis of the surgical site), a regional pedicled flap (i.e., pericranial or temporoparietal fascia flap) or a soft tissue free flap (e.g., anterolateral thigh, latissimus dorsi or rectus abdominis free flaps) could be indicated [105,106,107,108,109,110,111].

### 4.6. Surgery: Reconstruction after Resection of a Maxillary/Frontal Sinus SCC

A complex reconstruction is required after a large surgical resection involving the maxillary and/or frontal bone; the maxillary and frontal bones are centerpiece of the middle and upper face, respectively, and their resection has aesthetical, functional (i.e., oculopalpebral, speaking-, and swallowing-related), and vital (i.e., protection of the skull base, intracranial content, and vessels from microbes and desiccation) implications. Moreover, sinonasal malignancies most often require a multimodal management, with a high rate of patients sent for adjuvant treatments, [112,113,114] whose toxicity can be amplified in case of suboptimal reconstruction. Reconstruction of midfacial defects created through resection of an SNSCC constitutes a complex and technical challenge, and preoperative planning is paramount [115,116]. Locoregional, free flaps, and/or obturator prostheses should all be considered when planning reconstruction, whose complexity also needs to be weighed in view of a patient’s general conditions, comorbidities, ability to tolerate long procedures, and prognosis. A small defect of the hard palate can be repaired with local/regional flap (e.g., palatal mucoperiosteal island flap, temporalis muscle flap, temporoparietal fascia flap) or obturator prostheses. The gold standard for complex and large midfacial defects reconstruction consists of bone-containing free flaps (i.e., scapular, fibular, iliac crest free flaps). Other options include soft tissue free flaps (i.e., anterolateral thigh, rectus abdominis, radial forearm, latissimus dorsi flaps) or, in very selected cases or unfit patients, regional flaps (e.g., temporalis muscle, temporoparietal fascia flap) [117,118].

Owing to the rarity of SCC arising from the frontal sinus, [119] principles of reconstruction of frontal bone defects resulting from oncologic ablations are less univocally defined. Cranioplasty through titanium, ceramic, or plastic implants is among the most frequently employed strategies, and most invariably requires a two-stage surgery [120]. Autogenous calvarial grafts [121] or free flaps [122,123,124] have also been described to reconstruct defects of the frontal bone.

As a final remark, tissue bioengineering is expected to provide game-changing possibilities in the field of craniomaxillofacial reconstruction, but, to date, most reported experiences are anecdotal and applied to nononcologic scenarios [125].

### 4.7. Surgery: Neck Dissection

Involvement of cervical lymph nodes is considered one of the most significant prognostic factors in HNSCC. Clinical evidence of nodal metastases was reported to be around 13% in the largest series of nasal cavity and paranasal sinus SCCs [20]. The percentage increases to 19% for paranasal sinuses and to 21% for the maxillary sinus alone. In a meta-analysis of 1283 patients with SNSCCs, the initial nodal involvement rate for nasal cavity and maxillary sinus SCC was 9.3% and 20.7%, respectively [126]. In SCCs arising from the nasal cavity, a significantly higher rate of nodal involvement was found for primary tumors of 2 cm or larger and T4. Neither high-grade nor nasopharyngeal involvement significantly increased the risk of nodal involvement. For maxillary sinus SCCs, T2-or-higher tumors were associated with a higher rate of nodal metastasis, whereas size had no impact. Interestingly, Doescher et al. observed that only EBV-positive SNSCCs developed nodal metastasis in a series of 44 SNSCCs [13]. Based on these findings, it appears reasonable to consider maxillary sinus SCC as a different entity compared with nasoethmoidal SCC, at least with regards to the management of the neck.

SNSCC patients with clinically positive neck should receive a neck dissection as a part of surgery. There are no data to support one type of neck dissection over the others, therefore selective, comprehensive, or radical neck dissection have been alternatively advocated [127,128]. To the best of the authors’ knowledge, literature on definitive nonsurgical treatment of a neck bearing clinically appreciable metastasis is scarce and fragmented [129].

On the other hand, there is no consensus on how to manage the neck in patients affected by cN0 SNSCCs. In a large series of patients with maxillary sinus SCCs published by Cantù et al., the rate of neck metastases, either at presentation or after treatment, was much higher in patients with T2 tumors than in those with T3, T4a, or T4b tumors, differently from any other HN cancer [128]. Based on their findings, the authors concluded that elective neck treatment is not indicated in patients with T3N0 and T4N0 tumors, whilst it may be considered in T2N0 SCCs. Possible explanations for this countertrended statement are: (1) the inclusion of both primary and recurrent tumors in the same series; the latter category might have included patients with advanced recurrent tumors previously irradiated on the neck, with a possible response of clinically unnoticed nodal metastases. This selection bias may have increased the number of rcT3N0 and rcT4N0, with a consequently higher ratio of cT2N+ SNSCCs; (2) maxillary sinus SCCs invading the oral cavity were classified as T2 if not extended to other anatomical sites determining up-classification to T3 or T4. This pattern of invasion brings the tumor into an area with a highly represented lymphatic network, with a consequent increased risk of lymphatic spread. This hypothesis is supported by the findings published by Ferrari et al. in their series of 138 maxillary sinus cancers [113]; they found that the incidence of nodal metastasis was significantly associated with the presence of lymphovascular invasion, which, in turn, was higher when inferior structures (i.e., hard and soft palate, alveolar process, and buccinator muscle) were involved. Since the invasions of these structures configures a cT2 category but does not exclude a possible more advanced stage tumor, it seems more reasonable to suggest an elective treatment of the neck based on the inferior extension of a maxillary cancer rather than on its T category. Moreover, risk of nodal occult metastasis most probably depends upon tumor biology rather than merely on local extension and T classification [13].

Berger et al. [130] recently summed up the findings from four retrospective cohort studies [126,128,131,132] and one review [133] on elective neck dissection in cN0 SCCs of the maxillary sinus, stating that strong recommendations cannot be made for upfront elective neck dissection, as evidence and conclusions are heterogenous and there is a lack of prospective trials. Overall, they concluded that elective neck dissection or treatment of the neck should be considered for higher T categories (i.e., cT3-4), or when the neck is accessed to harvest the recipient vessels for a microvascular reconstruction. In a cohort of 124 patients affected by cN0 paranasal sinuses cancer (both SCCs and non-SCCs), Lee et al. found that elective neck treatment has no benefit on survival probability and failure patterns, regardless of the neck treatment modality employed [134]. In their study, Crawford et al. analyzed a group of 1220 patients affected by SNSCC. They did not find any statistically significant difference in OS between patients who underwent elective neck dissection and those who did not, with a global prevalence of occult metastases of 12.7%. The reported results were confirmed when the cohort was filtered to consider only maxillary sinus SCCs [135]. Nonetheless, no information about nodal recurrence-free survival was given. When analyzing outcomes related to neck treatment in SNSCCs, one should consider a number of potential confounders such as: (1) most cases are treated with a combination of surgery and RT(-ChT); (2) involvement of surgical margins is higher than in most other HN sites due to the close proximity to vital neurovascular structures; (3) the development of distant metastases via hematogenous spread is slightly higher compared to regional metastases [136]. Therefore, the scarce effect of elective neck dissection on oncological outcomes could be explained by the presence of more relevant negative prognosticators that might obscure the long-term benefit on regional control. Also, any survival benefit conferred by elective neck dissection could be eclipsed by the efficacy of RT(-ChT) electively delivered on the nodal basin in nondissected, cN0 patients.

Based on personal experience and data available in the literature, it is the authors’ opinion that neck dissection should be reasonably indicated in case of clinically appreciable nodal metastasis (i.e., comprehensive neck dissection) or when harvesting of recipient vessels is required for reconstruction purposes (i.e., selective neck dissection including levels IB to III) (Figure 4).

### 4.8. Adjuvant Treatment

Complete tumor removal followed by adjuvant intensity-modulated RT (IMRT) is the current mainstay of treatment for advanced SNSCCs (pT3-4) and in most intermediate tumors (pT2). SNSCCs classified as pT1 are anecdotal and, therefore, there are no data to support the use of unimodal vs. bimodal treatment. Some histological subtypes of SNSCC (i.e., HPV-related multiphenotypic sinonasal carcinoma and NKSCC) might have pronounced radiosensitivity [1].

IMRT is an advanced external-beam, photon-based RT that is most beneficial for complex and irregularly shaped target lesions located adjacent to critical organs at risk [137]. The relatively steep dose gradients achievable with IMRT helps the radiation oncologist in delineating the following volumes of treatment [138]: original insertional area of the tumor and microscopically involved margins are included in the clinical tumor volume (CTV) and targeted with 66 Gy (i.e., CTV_66_); areas at high risk for microscopic tumor invasion based on initial gross tumor extension and surgical violation are included within the CTV receiving 60 Gy (i.e., CTV_60_). A 0–5-mm margin is added to both CTV_66_ and CTV_60_ to delineate the planning target volumes (i.e., PTV_66_ and PTV_60_). Dose per fraction is commonly kept at 2 Gy. Areas that were both spared by surgery and deemed at low risk for microscopic disease should receive 54 Gy in 1.8 Gy-fractions (i.e., PTV_54_). In cN0 tumors for which elective neck irradiation is indicated, PTV_54_ includes also lymph nodal levels I to IV; level V should be included in case of nasoethmoidal or nasopharyngeal-involving SNSCC; retropharyngeal nodes are included in case of nasopharyngeal-involving SNSCC, or in case of nodal metastases from a nasoethmoidal SCC. The contralateral nodal basin is included in the radiation field based on a case-by-case evaluation.

Intensity-modulated particle therapy (IMPT) (i.e., protons and ^12^C-carbon ions) provides several advantages compared to photons-based RT, including the ability to design a sharp dose gradient owing to the so-called “Bragg peak”, increased radiobiological effectiveness thanks to high linear energy transfer, and relative independence of tissue oxygenation [139]. These physical and biophysical advantages translated into clinical benefits in such a complex district as the HN, with special reference to the skull base. There are several emerging proofs that IMPT provides satisfactory oncologic results in several sinonasal cancers, among which SNSCC [140,141]. Since IMPT is likely to play an important role as experience accrues [142,143], adjuvant proton therapy can be indicated in selected SNSCCs to reduce the risk of toxicity to adjacent critical structures, such as the optic system. In a series of 54 patients with stage III–IV SNSCC, Russo et al. investigated the long-term outcomes of proton therapy. In 37 cases (69%), the treatment was delivered after surgery and, in 24 patients (44%), in combination with either induction, concurrent, or adjuvant ChT. Elective neck irradiation was also performed in 74% of cases. They reported satisfactory locoregional control and OS with acceptable toxicity events (17% grade 3, 11% grade 4, 0% grade 5) [144].

The addition of platinum-based, concurrent ChT in the adjuvant setting is classically indicated in case of close or positive margins, or in the presence of other high-risk features, such as nodal metastasis with extranodal extension, perineural invasion or lymphovascular invasion [16]. As for other HNSCCs, the dose of cisplatin can be 100 mg/m^2^ every 3 weeks or 30–50 mg/m^2^ in weekly administration [145,146,147]. In case of neural, renal, or acoustic toxicity, replacement with carboplatin can be considered [148].

Timely co-ordination between surgeons and radiation oncologists is strongly recommended when a patient affected by SNSCC has to undergo a multimodal treatment. In fact, it has been recently demonstrated that surgery-to-radiation interval should be kept within 61 days to avoid outcomes worsening [84].

The rarity of SNSCC prevented any randomized trials investigating the efficacy of different adjuvant treatment options, and no large-scale controlled studies have been performed yet.

### 4.9. Definitive Photon-Based (Chemo)Radiation

Although surgery followed by adjuvant RT(-ChT) has been associated with increased survival outcomes in SNSCCs, [69] definitive (chemo)radiation may be considered in patients who refuse upfront surgery, for unresectable tumors, or when the surgical procedure required is deemed too invasive for a given subject [149]. In the majority of cases, RT is administered in the form of IMRT, whilst ChT is generally platinum-based and follows the same schedule described for the adjuvant setting. In a large series published by Robin et al., OS did not differ between definitive chemoradiation and surgery alone; on the other hand, definitive RT-ChT was associated with worse oncological results if compared to neoadjuvant therapy-based protocols and trimodal strategies [69]. By contrast, Park et al. found no significant difference in terms of local progression-free, regional progression-free, distant metastasis-free, and OS comparing definitive RT with surgery followed by adjuvant RT in their series of 73 patients affected by SNSCC [150]. Accordingly, they suggest definitive RT as a valuable substitute to surgically based protocols if a careful follow-up and adequate salvage options can be ensured. Qiu et al. compared definitive bioradiotherapy (i.e., RT combined with Cetuximab 400 mg/m^2^ the first dose and 250 mg/m^2^ with weekly administration thereafter) with RT alone [151]. They demonstrated that both objective response rate (77.4% versus 45.6%, respectively), progression-free survival (19.5 versus 13.8 months, respectively), and OS (26.6 versus 18.9 months, respectively) were higher in patients treated with bioradiotherapy.

Overall, sending a patient affected by SNSCC for definitive photon-based (chemo)radiation seems a less-than-optimal treatment path, which should be currently reserved to cases for which complete surgical resection is deemed unfeasible or unacceptably invasive. However, the marked and often unpredictable radiosensitivity displayed by some sinonasal cancers suggests that identification of reliable biomarkers to select SNSCCs which will respond to RT(-ChT) should be one among the several unmet aims in the field of sinonasal oncological research.

### 4.10. Definitive Particle (Chemo)Radiation

In view of their physical and biophysical properties, charged particle beams are ideally suited for dose escalation in complex anatomical sites located close to critical and radiosensitive organs [139]. The dosimetry advantage provided by IMPT over IMRT in HN tumors was recently demonstrated by Nguyen et al. [152]. IMPT is thereby considered as the gold standard treatment for skull base radioresistant tumors, such as chordomas and chondrosarcomas, which need to be irradiated with a high dose [153,154,155]. The largest series documenting the use of definitive IMPT in patients affected by SNSCCs was published by Toyomasu et al. in 2018 [156]. They reported a series of 59 patients, for whom IMPT conferred oncologic results that were consistent with those of other treatment strategies, with 5-year OS, progression-free survival, and local control rates of 41.6%, 34.7%, and 50.4%, respectively, yet with a non-negligible rate of late toxicity (rate of grade-3-or-higher toxicity: 22%). Recently, IMPT was also reported as a valuable strategy of radiation-based organ preservation treatment in a small cohort of 11 patients who refused upfront total rhinectomy for nasal cavity SCCs [157]. IMPT was administered in alternative to IMRT, and in combination with concurrent chemotherapy, with a 2-year rhinectomy-free survival, OS, and recurrence-free survival of 88%, 100%, and 75%, respectively. However, rhinectomy is often indicated for SCCs arising from the nasal vestibule, which have remarkably distinct characteristics compared to SCCs arising from the nasal cavity and paranasal sinuses (e.g., higher propensity towards nodal metastases, etiological relationship with tobacco smoking). Therefore, these results should not be merely translated to all SNSCCs.

A registered clinical trial is currently comparing definitive IMPT with IMRT for sinonasal malignancies (ClinicalTrials.gov (accessed on 1 June 2021) Identifier: NCT01586767), whereas another trial is investigating the role of R2 endoscopic surgery followed by IMPT in patients affected by unresectable sinonasal cancer (ClinicalTrials.gov (accessed on 1 June 2021) Identifier: NCT03274414).

### 4.11. Neck Irradiation

Irradiation is one of the possible treatment modalities of the neck. Many patients affected by SNSCC receive RT as part of their primary multimodal treatment, and the inclusion of the neck in the radiation fields is commonly reported.

Le et al. reported their experience with 97 patients affected by maxillary sinus malignancies, out of which 58 were SCCs [129]. Among 36 patients who received neck irradiation, 25 were cN0. They found a statistically significant difference in terms of nodal recurrence between the irradiated and the nonirradiated neck patients, with rates of 0% and 20%, respectively. In their series, they also found that SCC was associated with a higher incidence of initial nodal involvement and nodal relapse as compared to other histologies. All patients with nodal involvement had a T3–4 tumor, which led the authors to recommend elective neck irradiation in T3–4 SCC of the maxillary sinus. A similar recommendation came from a review published by Takes et al., in which they suggest that the regional lymphatics should be treated electively in all T3–4 cN0 maxillary sinus SCCs, whilst neck treatment of T2 tumors should be evaluated on a case-by-case basis [158].

## 5. Follow-Up, Treatment of Recurrences, and Palliation

As for all cancers, the main aims of follow-up after completion of locoregional therapy for SNSCC are early identification of recurrence and diagnosis and management of treatment complications and toxicity.

### 5.1. Complications and Sequelae after Locoregional Therapy

Complications after locoregional therapy for SNSCC are non-negligible and can substantially affect the quality of life of patients. Data available in the literature are not strictly related to SCC as morbidity is supposed to be treatment-related rather than histology-dependent. According to the largest series available, complication rate after craniofacial resection and open maxillectomy is as high as 36.3% and 64.9%, respectively [159,160]. Mortality of craniofacial resection has been reported being 4.7% in an international collaborative study on 1193 patients [159]. However, advances in surgical technique and technology that occurred over the last decades have remarkably reduced the morbidity and mortality of treatment [97,161,162,163]. Similarly, advances in the field of RT, such as intensity modulation and particle therapy, have substantially contributed to reduce the burden of post-treatment sequelae [164,165], so that long-term quality of life of sinonasal cancer survivors currently approaches that of the general population [162,166]. The main complications after treatment of SNSCC include wound, ocular, neurological, oral, and systemic complications, whose detailed discussion goes beyond the aim of the present review. Interestingly, recent evidence suggests that neuropsychological complications are more represented than previously thought. Lee et al. recently published an NCD study on 6760 subjects, demonstrating that prevalence of mental health disorders increases from 22 to 31% following diagnosis of skull base cancer, with depression and anxiety being the most relevant [167]. Interestingly, all treatment modalities showed an independent role in increasing the risk of mental disorders: odds ratios (OR) of RT, surgery, and ChT were 2.14, 1.82, and 1.63 in a multivariable model, respectively. Female gender (OR = 1.38), history of smoking (OR = 1.52), and alcohol dependance (OR = 3.18) resulted as patient-related independent predictors of mental disorder development. Sharma et al. recently demonstrated that long-term brain toxicity is not negligible in patients receiving RT for sinonasal cancer, with 37.0% of patients experiencing a clinically significant neurocognitive impairment [168].

Overall, successful management of morbidity following treatment of SNSCC is a challenge and, according to the authors’ opinion, it is best achieved within a multidisciplinary frame.

### 5.2. Clinical and Radiological Follow-Up Evaluations

Post-treatment imaging plays a critical role in follow-up of patients, as most areas to be checked to exclude a locoregional relapse are unreachable by endoscopy and/or clinical examination. The general recommendations on timing and type of examination to be adopted are reported in the NCCN guidelines on management of HN cancers [90,169]. However, it must be emphasized that evidence determining the adequate follow-up schedule of patients treated for SNSCC is particularly frail. A baseline MRI examination, currently recommended at 3–4 months post-treatment for HN cancer, has the twofold role of objectively measuring the response to treatment and providing a comparison for all future exams. A locoregional imaging is then recommended every 4 months during the first year, every 6 months from the second to the fifth year, and yearly thereafter. Regarding PET/CT, a time interval of 10–12 weeks after completion of RT is generally recommended to avoid false positive results due to early post-treatment changes, free tissue transfer, or inflammation in the primary site or regional lymph nodes; [170] interestingly, sinonasal malignancies have shown a longer period of hypermetabolism compared to other HN cancers [171].

### 5.3. Management of Locoregional Recurrences

In sinonasal malignancies, local recurrence is the main type of treatment failure, since regional and distant metastasis are infrequent [172]. The rate of local and regional recurrences in patients treated with surgery and adjuvant RT for a sinonasal malignancy is reported to range between 40 and 80% of cases [159,173,174,175], with a vast variability according to histology, stage, and other risk factors. Most of the SNSCC recurrences occur within 5 years following primary treatment [176].

Treatment of recurrences should be discussed in a multidisciplinary setting, with surgery currently representing the mainstay for resectable tumors. Owing to the rarity of recurrent SNSCCs, data and related thoughts should be extrapolated from studies including several histologies. Kaplan et al. analyzed 42 patients who underwent curative-intended surgery, with or without adjuvant therapy, for locally recurrent sinonasal malignancies [177]. They reported an average disease-free interval of 27 months and a 5-year OS rate of 47.6%. Based on their observations, the authors classified recurrent SNSCC within the cluster of “aggressive histologies” (i.e., SCC, adenocarcinoma, SNUC, mucosal melanoma, NECs, and sarcomas). They found that aggressive histology, poorly differentiated tumors, and extension beyond the nasoethmoidal box were negative prognostic factors. Therefore, the authors concluded that nasoethmoidal relapsing “aggressive histologies”, with/without invasion of the orbit and/or skull base, should be sent for surgery based on case-by-case evaluation, whereas non-nasoethmoidal recurrences should not be referred to surgery with curative intent. Despite the chance to cure some patients affected by recurrent SNSCC being limited, owing to the high risk of distant metastasis, surgery and/or other locoregional treatments with additional/equal control probability should be considered whenever the lesion is resectable with an acceptable morbidity, since uncontrolled local progression of sinonasal malignancies is associated with a dismal quality of residual life [176,178]. Nonradical surgery with palliative intent was also reported as a potential strategy for uncurable patients. Tabaee et al. reported on 6 patients who received endoscopic surgery to relieve symptoms such as nasal obstruction and epistaxis [179]. Five (83.3%) patients were local symptom-free for a time span ranging from 1 to 6 months. Mean survival after surgery was around 6 months. An NCD study on 380 patients showed that only surgery-including active palliation strategies were associated with a significantly increased OS compared to best supportive care, independently of age, comorbidity, gender, stage, histology, and race [180].

Reirradiation (reRT) is another resource of the therapeutic armamentarium for recurrent sinonasal cancers. ReRT can be indicated as either standalone or adjuvant modality following surgery. Currently, benefits of reRT of recurrent sinonasal cancer in terms of survival and locoregional control is unclear, whereas the potential sequelae, such as osteoradionecrosis and vascular blow-out, are well known and particularly feared [181,182]. A reRT dose that can be safely delivered is frequently restricted by close proximity to critical neurovascular structures and/or by the location of the recurrence in a previously irradiated area [183,184]. IMPT might partially overcome the limitations faced when reRT is indicated [185,186,187].

Stereotactic RT should also be considered in the palliative setting. In particular, stereotactic RT of recurrent, unresectable tumors determining trigeminal neuralgia has proved to be effective in decreasing patient-reported pain and opioid requirement [188].

### 5.4. Management of Distant Recurrences

ChT can be indicated for locoregional recurrences deemed ineligible to locoregional treatment and/or in case of distant metastasis. Drugs used in the palliative setting are similar to those mentioned for neoadjuvant ChT; the choice of combination therapy vs. monotherapy and dose depend on clinical conditions of the patient [189]. The role of ChT in the salvage and palliative setting of sinonasal malignancies is not supported by sound evidence, mostly due to the rarity of these tumors. However, there is evidence that recurrent/metastatic sinonasal tumors showing objective response to palliative ChT are associated with remarkably longer survival (29.2 months) compared to those with progression of disease (4.4 months) [178]. Orlandi et al. reported that palliative ChT was either platinum-, anthracycline-, taxane-, and/or alkylating-agent-based (e.g., temozolomide or ifosfamide). Most patients received polychemotherapy [178].

Systemic immunotherapy with nivolumab or pembrolizumab is in the process of being possibly entitled as alternative “first line treatment” for recurrent/metastatic HNSCC. However, very little information on the potential effect of immunotherapy on sinonasal tumors is currently available. Riobello et al. demonstrated that membranous expression of PD-L1 is observed in 34% of the tumor cells and in 45% of immune infiltrating cells of SNSCC, thus alluding to a potential benefit from immunotherapy in selected cases [190]. Overall, the role of immunotherapy in sinonasal and skull base malignancies is still far from being fully elucidated, as the treatment reports of patients with these cancers are limited. Similarly, as recently highlighted by Licitra et al., there is paucity of data on biotherapy on SNSCC, as sinonasal cancers are frequently excluded from trials testing targeted therapy [189].

Metastasectomy should also be considered in well selected patients. Location and number of distant metastases, and distant metastasis-free interval following presentation are the main factors to take into account. In sinonasal histologies with well-known quick kinetics, such as SNSCC, the threshold of the distant metastasis-free interval to stratify prognosis was identified earlier than for notoriously slower progressing tumors, such as adenoid cystic carcinoma [191]; Lu et al. found that a 12 month or longer distant metastasis-free interval is significantly associated with a better outcome in patients who undergo pulmonary metastasectomy for metastatic HNSCC [192].

Adequate palliation may require different modalities and, therefore, the same actors playing a major role in curative treatments are likely to be somehow involved also in noncurative care. Given the different aim of palliation as opposed to curative treatment, complications and toxicity are, by far, less acceptable, and any treatment modality should be tailored accordingly.

## 6. Conclusions

Rarity, biological diversity, and heterogeneity of management are the main factors limiting the understanding of SNSCC. These aspects resulted in a limited level of evidence, which does not overcome the threshold of 2–3 (i.e., systematic review of nonrandomized clinical trials); given the majority of current thoughts on SNSCC are based on level 3-to-5 evidence, recommendations resulting from scientific data should be interpreted cautiously. The present review has highlighted a number of recent refinements in the molecular definition of sinonasal epithelial cancers. Multimodal imaging is essential to reliably stage the local, regional, and systemic extension of the tumor. Surgical ablation is the mainstay of treatment. Neoadjuvant ChT is currently indicated with different aims, including orbit preservation, stratification of prognosis, chemoselection of locoregional treatment, and reduction of distant metastases. Neoadjuvant ChT-RT has been advocated to increase the chances of sparing the orbit and improving the margin status in patients receiving surgery. IMRT, possibly combined with concomitant ChT, plays an essential role in the curative-intended treatment of the large majority of SNSCCs. The advent of IMPT has been providing promising results in the field of sinonasal and skull base cancer treatment and is expected to also be beneficial for SNSCCs. Although the management of the neck is far from being standardized, an algorithm is hereby presented based on available evidence and authors’ personal experience (Figure 4). Management of recurrent and uncurable SNSCCs is not standardized and deserves dedicated research.

## Figures and Tables

**Figure 1 cancers-13-02835-f001:**
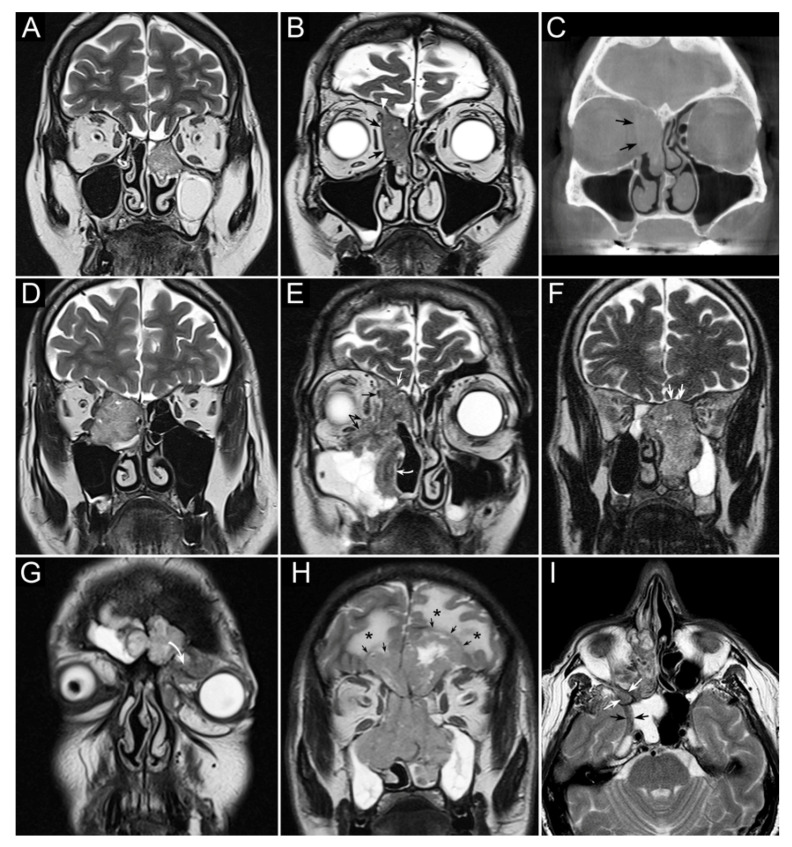
Panel summarizing several patterns of local extension of nasoethmoidal squamous cell carcinoma (SCC) and other epithelial cancers. (**A**) Left nasoethmoidal SCC with neuroendocrine features, limited to the nasoethmoidal compartment. (**B**,**C**) Non-keratinizing SCC of the right nasoethmoidal compartment determining resorption of the lamina papyracea (arrows in **C**), with no frank transgression of the periorbit. A T2-hypointense line is visible between the tumor and extraconal fat (arrows in **B**). The supraorbital ethmoidal recess is filled by dehydrated secretions (arrowhead in **B**). (**D**) Orbit-encroaching right nasoethmoidal SMARCB1/INI1-deficient carcinoma. A T2-hypointense line separating the tumor from the extraconal fat cannot be clearly demonstrated. (**E**) Right nasoethmoidal poorly differentiated SCC invading the extraconal fat (black arrows), anterior skull base (white arrow), and medial maxillary wall (curved white arrow). (**F**) Left nasoethmoidal adenosquamous carcinoma determining cranial displacement of the T2-hypointense thin layer made up by the ethmoidal-sphenoidal roof and overlying dura mater (white arrows). (**G**) Left nasoethmoidal SMARCB1/INI1-deficient carcinoma involving both frontal sinuses and invading the orbital cavity through the floor of the left frontal sinus (curved arrow). (**H**) Non-intestinal-type adenocarcinoma of the nasoethmoidal box with massive transcranial extension (arrows). Brain edema (asterisks) suggests invasion of the cerebral parenchyma. (**I**) Poorly differentiated SCC of the right nasoethmoidal compartment, with involvement of the inferior orbital fissure (white arrows) and macroscopic perineural extension along a thickened maxillary nerve to reach the Meckel’s cave (black arrows).

**Figure 2 cancers-13-02835-f002:**
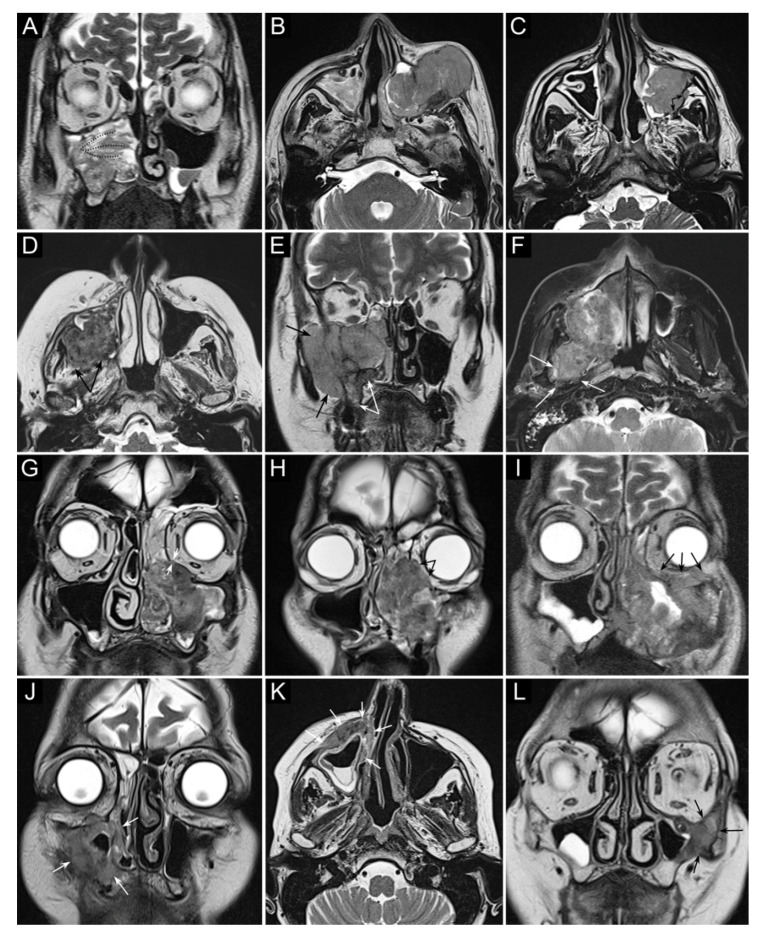
Panel summarizing several patterns of local extension of maxillary squamous cell carcinoma (SCC) and other epithelial cancers. (**A**) SCC ex inverted papilloma limited to the right maxillary sinus. A quite regular striated appearance indicates the papilloma (dotted curved lines). (**B**) Poorly differentiated SCC of the left maxillary sinus with massive infiltration of premaxillary tissues. (**C**) Spindle-cell SCC of the left maxillary sinus with initial infiltration of the fat pad located within the infratemporal fossa (arrows). (**D**) Moderately differentiated SCC of the right maxillary sinus with massive infiltration of the infratemporal fossa (arrows). (**E**) Neuroendocrine carcinoma of the right maxillary sinus infiltrating the ipsilateral nasoethmoidal compartment and alveolar process (white arrows), and massively invading the infratemporal and temporal fossae (black arrows). (**F**) Poorly differentiated SCC of the right maxillary sinus invading the infratemporal fossa, medial and lateral pterygoid muscle, and the upper parapharyngeal space (arrows). (**G**) Poorly differentiated SCC of the left maxillary sinus with initial orbital encroachment; the T2-hypointense line separating the tumor from the extraconal fat is partially interrupted, suggesting initial invasion of the extraconal compartment (arrows). (**H**) Moderately differentiated, keratinizing SCC of the left maxillary sinus with orbital involvement; a T2-hypointense line separating the tumor from the extraconal fat cannot be depicted (arrows). (**I**) Poorly differentiated SCC of the left maxillary sinus with massive invasion of the extraconal fat (arrows). (**J**–**L**) Three examples of SCC of the maxillary sinus with permeative bone invasion, which consists of cancer progressing within bone structures while partially maintaining their initial shape (arrows).

**Figure 3 cancers-13-02835-f003:**
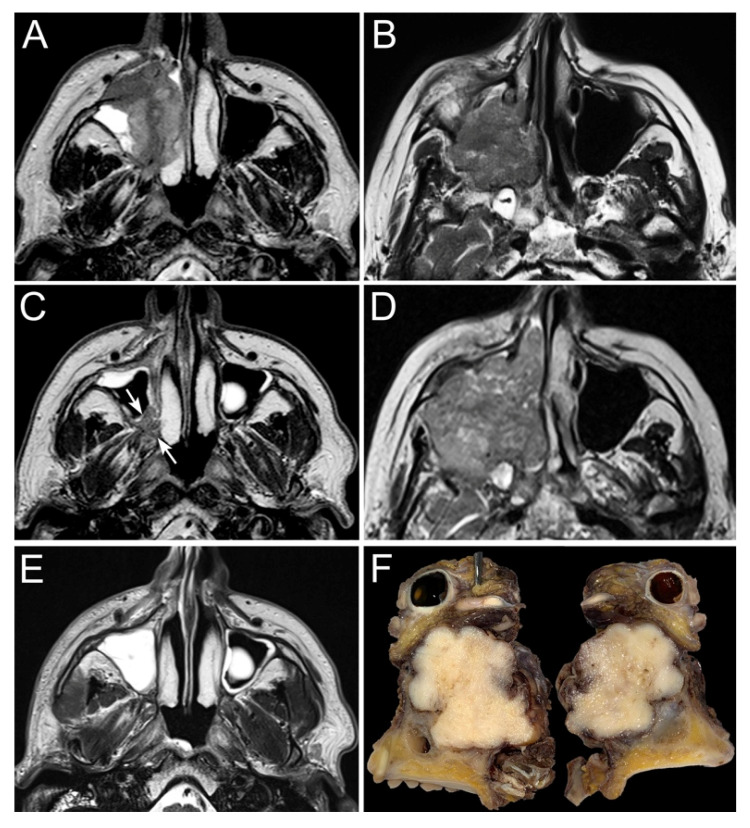
The panel shows two examples of locally advanced poorly differentiated squamous cell carcinoma of the right maxillary sinus (**A**,**B**). Both patients underwent neoadjuvant chemotherapy (3 cycles) with taxane, cisplatin, and 5-fluorouracile, which resulted in a heterogenous response: the first tumor (**C**) showed a partial response (arrows), whereas the second one (**D**) progressed throughout chemotherapy. Based on the response to chemotherapy, the first patient was treated through intensity-modulated proton therapy combined with concomitant platinum-based chemotherapy, which led to a complete response with no relevant toxicity (**E**). The second patient was instead operated on through a total maxillectomy extended to the infratemporal fossa and upper parapharyngeal space, orbital exenteration (**F**), anterior and middle craniectomy with dural resection, skull base multilayer graft-based reconstruction, and anterolateral thigh free flap.

**Figure 4 cancers-13-02835-f004:**
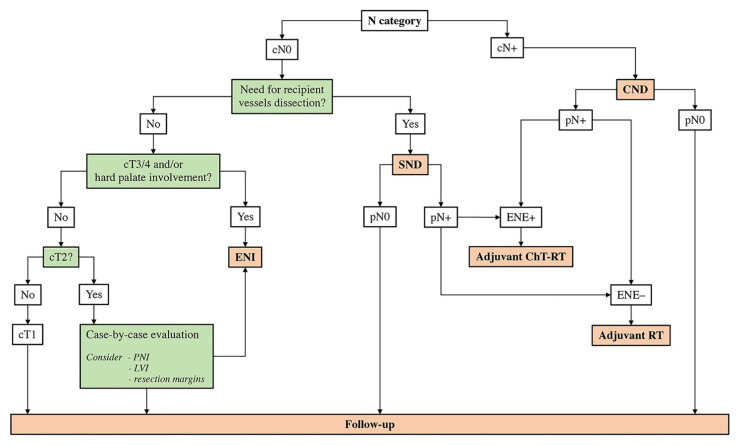
Flowchart summarizing the neck management strategy proposed based on available literature and authors’ experience. CND, comprehensive neck dissection; ChT, chemotherapy; ENE, extranodal extension; ENI, elective neck irradiation; LVI, lymphovascular invasion; N, nodal; PNI, perineural invasion; RT, radiotherapy; SND, selective neck dissection.

**Table 1 cancers-13-02835-t001:** Table summarizing the goals of neoadjuvant chemotherapy (ChT) and chemoradiation (ChT-RT) alongside supporting evidence. A, adriamycin; C, cetuximab; CR, complete response; E, etoposide; F, 5-fluorouracile; I, ifosfamide; N.A., not available; P, cisplatin; T, taxane. * Concurrent to RT; ** Hirawaka et al. [58] used different evaluation criteria to measure response to treatment, other authors used RECIST (Response Evaluation Criteria In Solid Tumors).

Rationale	Supporting Evidence	Type of Neoadjuvant Treatment (Objective Response Rate)	ChT Regimens	Description	References
Orbit preservation	Yes	ChT (61.9–100%)	TPF	Neoadjuvant ChT is associated with a considerable chance of orbit preservation (range: 60.0–78.8%) in patients for whom orbital ablation would be indicated in case of upfront surgery.	Ock et al. [63] Hanna et al. [62] Chen et al. [64] Turri-Zanoni et al. [65] Ferrari et al. [49] Fernström et al. [66]
			TPI		
			TP		
			TF		
			PF		
		ChT-RT (57.1%; CR rate: 35.7–48.1%)	P *	Neoadjuvant ChT-RT reduces the rate of orbital ablation in patients with orbit-abutting SNSCC (range: 93–100%).	Amsbaugh et al. [67] Fernström et al. [66]
Stratification of prognosis	Yes	ChT (67.4–93.0% **)	TPF	A good response to neoadjuvant ChT is associated with a relatively favorable prognosis in terms of recurrence and survival.	Hanna et al. [62] Hirakawa et al. [68]
			TPI		
			TF		
			PF		
Improvement of margin status	Yes	ChT-RT (N.A.)	N.A.	Neoadjuvant ChT-RT reduces the rate of involved margins in patients with orbit-abutting SNSCC.	Robin et al. [69]
Chemoselection	No	ChT (56.9%)	TPF	Effectiveness of locoregional treatment (surgery vs. (ChT)-RT) does not relate to response to neoadjuvant ChT.	Abdelmeguid et al. [70]
			TPI		
			TP		
			TP-C		
Improvement of distant metastasis-free survival	No	ChT (56.9%)	TPF	Neoadjuvant ChT could reduce the risk of distant metastases; however, comparative evidence is lacking.	Bossi et al. [71] Abdelmeguid et al. [70]
			TPI		
			TP		
			TP-C		

**Table 2 cancers-13-02835-t002:** Table summarizing the potential extensions (clustered in vectors of growth) of sinonasal squamous cell carcinomas alongside the recommended resection strategy, based on available literature and authors’ experience.

Primary Subsite	Extension Vector	Involved Structures	Recommended Surgery
Nasoethmoidal	Anterior	Nasal bones, frontal process of the maxillary bone, external nose	Partial/total rhinectomy
	Lateral	Medial orbital bony wall, periorbit, extraconal fat (minimal), medial wall of the lacrimal sac	Orbit-sparing endoscopic resection
		Extraconal fat (non-minimal), ocular muscles, eye, preseptal structures, orbital apex, lateral wall of the lacrimal sac	Orbital exenteration/clearance
	Medial	Nasal septum	Endoscopic resection
	Posterior	Sphenoid sinus, nasopharynx, sella, clivus	Endoscopic resection
	Posterolateral	Pterygopalatine fossa, infratemporal fossa, upper parapharyngeal space	Endoscopic resection
	Cranial	Bony skull base (ethmoidal roof, cribriform plate), dura mater, falx cerebri (minimal), brain (minimal)	Endoscopic resection with transcranial craniectomy w/o subpial dissection
		Falx cerebri (non-minimal), brain (non-minimal)	Cranioendoscopic resection w/o subpial dissection
	Cranio-lateral	Orbital roof, supraorbital dura	Endoscopic resection or cranioendoscopic resection
	Caudal	Hard palate	Inferior maxillectomy
Maxillary	Anterior	Premaxillary periosteum, subcutaneous tissue, skin	Maxillectomy w/o resection of premaxillary skin and rhinectomy
	Lateral	Buccal space, masticatory space	Maxillectomy w/o coronoidectomy
	Medial	Medial maxillary wall, nasolacrimal duct, nasal septum	Maxillectomy w/o septectomy
	Posterior	Pterygopalatine fossa, pterygoid process, masticatory space, upper parapharyngeal space	Maxillectomy w/o endoscopic-assisted delineation of the posterior margin
	Cranial	Maxillary sinus lumen	Subtotal maxillectomy
		Orbital floor, periorbit, extraconal fat (minimal)	Total maxillectomy w/o resection of the periorbit/extraconal fat
		Extraconal fat (non-minimal), ocular muscles, eye, preseptal structures, orbital apex	Orbital exenteration/clearance
	Caudal	Buccal space, buccinator muscle, alveolar ridge/gum, hard palate	Maxillectomy
Frontal	Anterior	Anterior frontal plate, periosteum, subcutaneous tissue, skin	Riedel’s operation
	Posterior	Posterior frontal plate, dura, brain	Riedel’s operation with posterior frontal craniectomy
		Bony skull base (ethmoidal roof, olfactory cleft) and overlying dura	Cranioendoscopic resection
	Caudal	External nose	Partial/total rhinectomy
		Orbital roof, periorbit, extraconal fat (minimal)	Orbit-sparing Riedel’s operation
		Extraconal fat (non-minimal), ocular muscles, eye, preseptal structures	Riedel’s operation with orbital exenteration/clearance
Sphenoid	Anterior	Anterior sphenoidal wall	Endoscopic resection
	Lateral	Lateral sphenoidal wall	Endoscopic resection
	Posterior	Sella, clivus	Endoscopic resection w/o transnasal craniectomy
	Cranial	Mucosa, planum sphenoidale, dura	Endoscopic resection w/o transnasal craniectomy
	Caudal	Sphenoidal floor, nasopharynx	Endoscopic resection

## Data Availability

No new data were created or analyzed in this study. Data sharing is not applicable to this article.

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
