# Peer review of "Sinonasal Squamous Cell Carcinoma, a Narrative Reappraisal of the Current Evidence"

_cancers, 2021, doi:10.3390/cancers13112835_

Round 1

Reviewer 1 Report

Excellent review of the contemporary management of sinonasal SCC. The review is comprehensive, scholarly, and well presented in the manuscript. 

Author Response

We thank Reviewer 1 for taking the time to evaluate the narrative review and for the appreciation.

Reviewer 2 Report

Ferrari et al. performed a comprehensive literature review about sinonasal SCC. The epidemiology, biology, pathology, diagnosis, staging, treatment, and post treatment surveillance have been reviewed and discussed in detail. The manuscript are written well and clearly. The following questions are for the authors.

  1. In table 1, authors summarized several studies of neoadjuvant chemotherapy (ChT) and chemoradiation (ChT-RT) extensions in sinonasal SCC. It is better if authors can simultaneously put the chemotherapy regimen and response rate in table 1. It makes readers understand easily.

  1. Neoadjuvant chemoradiotherapy or definitive chemoradiotherapy are commonly applied in advanced sinonasal SCC. Does different radiation dose have any impact on treatment effect and outcome? If references are available, please discuss the impact of radiotherapy dose.

  1. It is well-known that radiotherapy can increase surgical morbidity, even mortality in some cancers. For patients with advanced sinonasal SCC, does neoadjuvant chemotherapy or neoadjuvant chemoradiotherapy have different surgical morbidity or mortality?

  1. Some studies showed that some sinonasal cancer is EBV-related. It is well-known that EBV-related cancer is chemosensitive and radiosensitive disease. Please describe the association between sinonasal SCC with EBV if references are available.

  1. Are there any long-term noteworthy sequelae after surgery or radiotherapy for sinonasal SCC. Please discuss it.

  1. Some chemotherapy regimens can be described more clearly. For example, authors described that…there is evidence that recur-673 rent/metastatic sinonasal tumors showing objective response to palliative chemotherapy 674 are associated with remarkably longer survival (29.2 months) compared to those with pro-675 gression of disease (4.4 months) [160]….. It is better if authors can describe the chemotherapy regimen for this trial.

  1. Target therapy such as EGFR monocloncal antibody, cetuximab, or EGFR tyrosine kinase inhibitor, afatinib has been commonly applied in recurrent or metastatic head and neck squamous cell carcinoma. Are there any reports about EGFR monoclonal antibody or tyrosine kinase inhibitors in sinonasal SCC?

Author Response

Dear Reviewer,

Thank you for your comments, which led to substantially improved our manuscript in its revised version. Please find a point-by-point response below

Point-by-point response to Reviewer 2

Ferrari et al. performed a comprehensive literature review about sinonasal SCC. The epidemiology, biology, pathology, diagnosis, staging, treatment, and post treatment surveillance have been reviewed and discussed in detail. The manuscript are written well and clearly. The following questions are for the authors.

Thank you for appreciating our efforts.

  1. “In table 1, authors summarized several studies of neoadjuvant chemotherapy (ChT) and chemoradiation (ChT-RT) extensions in sinonasal SCC. It is better if authors can simultaneously put the chemotherapy regimen and response rate in table 1. It makes readers understand easily.

Thank you for your advice. As suggested, the chemotherapy regimen and objective response rate were added, when available.

2. “Neoadjuvant chemoradiotherapy or definitive chemoradiotherapy are commonly applied in advanced sinonasal SCC. Does different radiation dose have any impact on treatment effect and outcome? If references are available, please discuss the impact of radiotherapy dose.

Thank you for this input. However, neoadjuvant radiotherapy is commonly delivered with a dose of 50 Gy, whilst adjuvant RT as 66 Gy. To the best of authors’ knowledge, no comparative study is available to make firm conclusions on dose regimen to be preferred over those abovementioned. The only exception is a study by Kreppel et al., which has been added to the review to address your comment.

The following sentences have been added.

“Radiation dose is most commonly 50 Gy in 25 fractions.[56,57,68] Kreppel et al. did not find a significance difference in OS between 40 Gy and 50 Gy total RT dose.[69]

3. “It is well-known that radiotherapy can increase surgical morbidity, even mortality in some cancers. For patients with advanced sinonasal SCC, does neoadjuvant chemotherapy or neoadjuvant chemoradiotherapy have different surgical morbidity or mortality?

Thank you for highlighting this. To the best of authors’ knowledge, there are no data clearly demonstrating that postoperative complications increase following neoadjuvant (radio)chemotherapy for sinonasal squamous cell carcinoma. We reported a result approaching but not reaching significance on the rate of postoperative cerebrospinal fluid leak in patients who had received chemotherapy and or radiotherapy prior to surgery for nasoethmoidal cancer:

“Although one could hypothesize that neoadjuvant (ChT)-RT increases the risk of postoperative complications, sound data confirming this assumption are currently lacking. Mattavelli et al. found that previous ChT and/or RT increases the risk of postoperative cerebrospinal fluid (CSF) leak from 5.1% to 22.2%, though with no statistical significance.[70]

4. “Some studies showed that some sinonasal cancer is EBV-related. It is well-known that EBV-related cancer is chemosensitive and radiosensitive disease. Please describe the association between sinonasal SCC with EBV if references are available.

Thank you for this suggestion. Unfortunately, we were not able to find evidence that chemosensitivity and radiosensitivity can be inferred based on EBV status in sinonasal squamous cell carcinomas. However, based on your input, we found 2 interesting papers addressing the role of EBV in sinonasal squamous cell carcinomas. These studies (Doescher et al. and Nukpook et al.) showed that EBV infection prevalence in squamous cell carcinoma and nasal polyps is similar, but EBV-positive squamous cell carcinoma has a higher rate of nodal metastasis. Since we find these findings worth being mentioned in this narrative review, they have been cited as follows:

“In a similar fashion, Epstein-Barr virus (EBV) has been detected in a relevant proportion of SNSCCs.[7,8] However, since a similar proportion of EBV infection was observed in nasal polyps, its role in SNSCC carcinogenesis is questionable.[8]

“Interestingly, Doescher et al. observed that only EBV-positive SNSCCs developed nodal metastasis in a series of 44 SNSCCs.[7]

5. “Are there any long-term noteworthy sequelae after surgery or radiotherapy for sinonasal SCC. Please discuss it.

Thank you for underlining this aspect. The paragraph on follow-up after primary treatment has been divided in subparagraphs. A paragraph that briefly summarizes the main data and concepts on morbidity after treatment of sinonasal squamous cell carcinoma has been added.

5.1. Complications and sequelae after locoregional therapy

Complications after locoregional therapy for SNSCC are non-negligible and can substantially affect quality of life of patients. Data available in the literature are not strictly related to SCC as morbidity is supposed to be treatment-related rather than histology-dependent. According to the largest series available, complication rate after craniofacial resection and open maxillectomy is as high as 36.3% and 64.9%, respectively.[152,153] Mortality of craniofacial resection has been reported being 4.7% in an international collaborative study on 1193 patients.[152] However, advances in surgical technique and technology that occurred over the last decades have remarkably reduced the morbidity and mortality of treatment.[88,154–156] Similarly, advances in the field of radiotherapy such as intensity modulation and particle therapy have substantially contributed to reduce the burden of post-treatment sequelae,[157,158] so that long-term quality of life of sinonasal cancer survivors currently approaches that of the general population.[155,159] The main complications after treatment of SNSCC include wound, ocular, neurological, oral, and systemic complications, whose detailed discussion goes beyond the aim of the present review. Interestingly, recent evidence suggests that neuropsychological complications are more represented than previously thought. Lee et al. recently published a NCD study on 6760 subjects, demonstrating that prevalence of mental health disorders increases from 22% to 31% following diagnosis of skull base cancer, with depression and anxiety being the most relevant.[160] Interestingly, all treatment modalities showed an independent role in increasing the risk of mental disorders: odds ratios (OR) of radiotherapy, surgery, and chemotherapy were 2.14, 1.82, and 1.63 in a multivariable model, respectively. Female gender (OR=1.38), history of smoking (OR=1.52), and alcohol dependance (OR=3.18) resulted as patient-related independent predictors of mental disorder development. Sharma et al. recently demonstrated that long-term brain toxicity is not negligible in patients receiving RT for sinonasal cancer, with 37.0% patients experiencing a clinically significant neurocognitive impairment.[161]

Overall, successful management of morbidity following treatment of SNSCC is a challenge and, according to authors’ opinion, it is best achieved within a multidisciplinary frame.

6. “Some chemotherapy regimens can be described more clearly. For example, authors described that…there is evidence that recurrent/metastatic sinonasal tumors showing objective response to palliative chemotherapy are associated with remarkably longer survival (29.2 months) compared to those with progression of disease (4.4 months) [160]….. It is better if authors can describe the chemotherapy regimen for this trial.”

Thank you for raising this point. However, there is extreme paucity of data on palliative chemotherapy for sinonasal cancers. We extended the report of data on the reference that you highlighted as follows:

“Orlandi et al. reported that palliative ChT was either platinum-, anthracycline-based, taxane-, and/or alkylating agent-based (e.g., temozolomide or ifosfamide). Most patients received polychemotherapy.[173]

7. “Target therapy such as EGFR monocloncal antibody, cetuximab, or EGFR tyrosine kinase inhibitor, afatinib has been commonly applied in recurrent or metastatic head and neck squamous cell carcinoma. Are there any reports about EGFR monoclonal antibody or tyrosine kinase inhibitors in sinonasal SCC?”

Thank you for this valuable comment. Unfortunately, as reported by Licitra et al., sinonasal cancers are frequently excluded from targeted therapy trials, thus resulting in paucity of data on the effectiveness of biotherapy on these rare tumors. The fact that 8 patients of the series published by Abdelmeguid et al. received Cetuximab as part of neoadjuvant therapy was added. The study by Qiu et al., which compares treatments based on either radiotherapy and Cetuximab or radiotherapy alone, was added to the review, as well as the statement that little is known on biotherapy for sinonasal cancers.

Of note, 8 (6.5%) patients received also Cetuximab as part of the neoadjuvant therapy.

Qiu et al. compared definitive bioradiotherapy (i.e., RT combined with Cetuximab 400 mg/m2 the first dose and 250 mg/m2 with weekly administration thereafter) with RT alone. They demonstrated that both objective response rate (77.4% versus 45.6%, respectively), progression-free survival (19.5 versus 13.8 months, respectively), and OS (26.6 versus 18.9 months, respectively) were higher in patients treated with bioradiotherapy.

Similarly, as recently highlighted by Licitra et al., there is paucity of data on biotherapy on SNSCC, as sinonasal cancers are frequently excluded from trials testing targeted therapy.[184]

Reviewer 3 Report

This is a review article discussing current evidence on sinonasal squamous cell carcinoma. This is indeed a field in which concepts are evolving and new knowledge is emerging as new entities are identified and new treatment options are explored. Here are a few comments and suggestions:

1- The abstract mentions that a narrative literature review was performed (line 35) but that the paper also includes the authors' personal opinion on unsolved issues of this tumor (line 37). A review paper should highlight the areas where consistent high level evidence is present, and areas where conflicting reports or controversies exist. For the latter, nuanced discussion is included with possibly some opinions based on the authors experience. There is throughout the manuscript some overlap between higher level evidence and authors opinions which for readers who are not up to date on the literature related to sinonasal cancers, may be confusing. The authors should make it clear throughout the manuscript when an "expert opinion" is being inserted in the literature review when information is lacking or for controversies.

2- There absolutely is a relationship between tobacco exposure and sinonasal squamous cell carcinoma. The statement on lines 60 -61 is simply not correct. There absolutely is heterogeneous behavior and biological features with sinonasal squamous cell carcinomas but mentioning the overall between SCC, SNUC and SNEC is slightly confusing, as the manuscript is on SCC. Within the SCC, there are several entities that are not SNUC or SNEC and those are well described in the following paragraph. 

3-The information on the role of HPV in sinonasal squamous cell carcinoma is outdated. The references on this topic are old [ref 16-18]. Several high quality papers were published on the topic in 2020 and 2021 including in Head and Neck as well as Cancers. The authors should cite more recent references on this topic.

4- The staging paragraph needs the AJCC 8th Ed staging for sinonasal cancers and the modifications to T staging compared to the 7th Ed. Special mention should be made with respect to T4b / non surgical disease. This is important as the authors presents later in the manuscript case examples of what would be considered non surgical disease treated with surgery (including dural resection and repair)

5- The paragraph on neoadjuvant chemotherapy should include the ongoing multi institutional clinical trial EA3163 (https://ecog-acrin.org/clinical-trials/ea3163-educational-materials) 

6- Table 2 includes several categories of cases that would be considered non surgical for sinonasal squamous cell carcinomas with the corresponding surgery. This is more the author's experience than a literature review. Orbital apex, clivus, sella, falx cerebri, brain etc involvement by squamous cell carcinoma would be considered non surgical disease. Review of the current AJCC staging and NCCN guidelines would provide a more accurate view of the currently accepted surgical indications for sinonasal cancer. The NCCN guidelines are mentioned at the end of the manuscript (line 612) for follow up after treatment. There is merit in reviewing the guidelines throughout the manuscript as those are based on extensive literature review and current evidence.

7- Reconstruction should be rewritten in a smaller paragraph and included within the surgical treatment paragraph. There is nothing unique to squamous cell carcinoma. This is rather a section purely on surgical technique. The significant relevance of the reconstruction from an oncologic perspective is that failure of reconstruction could delay adjuvant treatment which is crucial to optimize the chances of cure in moderately advanced sinonasal cancer.

8- The section on adjuvant therapy is too long. The section on follow up and management of recurrent disease is too long.

Author Response

Dear Reviewer,

Thank you for your comments, which led to substantially improved our manuscript in its revised version. Please find a point-by-point response in the attachment.

Regards,

Marco Ferrari, MD

Point-by-point response to Reviewer 3

“This is a review article discussing current evidence on sinonasal squamous cell carcinoma. This is indeed a field in which concepts are evolving and new knowledge is emerging as new entities are identified and new treatment options are explored. Here are a few comments and suggestions:

  • The abstract mentions that a narrative literature review was performed (line 35) but that the paper also includes the authors' personal opinion on unsolved issues of this tumor (line 37). A review paper should highlight the areas where consistent high level evidence is present, and areas where conflicting reports or controversies exist. For the latter, nuanced discussion is included with possibly some opinions based on the authors experience. There is throughout the manuscript some overlap between higher level evidence and authors opinions which for readers who are not up to date on the literature related to sinonasal cancers, may be confusing. The authors should make it clear throughout the manuscript when an "expert opinion" is being inserted in the literature review when information is lacking or for controversies.”

Thank you for highlighting the importance of stating the level of evidence of the content of our review. It is well known that the level of evidence is particularly frail for any aspect concerning sinonasal cancer. We think that using the expression “authors’ opinion”, which we used when expressing our experience-based opinion on a particular aspect, could be sufficient to the reader to understand that a particular statement should be interpreted as no more than the authors’ thought.

To make it even clearer, the following sentences have been added to the first and last part of the manuscript, respectively:

“For those aspects lacking in evidence and consensus, the authors will report their experience-based opinion, yet recognizing it represents the lowest level of evidence in the scientific literature.”

“These aspects resulted in a limited level of evidence, which does not overcome the threshold of 2-3 (i.e., systematic review of non-randomized clinical trials); being the majority of current thoughts on SNSCC based on level 3-to-5 evidence, recommendations resulting from scientific data should be interpreted cautiously.”

  • “There absolutely is a relationship between tobacco exposure and sinonasal squamous cell carcinoma. The statement on lines 60 -61 is simply not correct. There absolutely is heterogeneous behavior and biological features with sinonasal squamous cell carcinomas but mentioning the overall between SCC, SNUC and SNEC is slightly confusing, as the manuscript is on SCC. Within the SCC, there are several entities that are not SNUC or SNEC and those are well described in the following paragraph.”

Thank you for raising this interesting debate. However, we humbly disagree on the fact that there is a certain etiological effect of tobacco smoking on development of sinonasal squamous cell carcinoma. We were not able to find clear and updated evidence on this aspect, while several papers questioned a potential role of tobacco in increasing the risk of sinonasal cancer. We were able to identify: 1) papers dating back to the 1980s observing a mild association between tobacco smoking and sinonasal cancer; 2) papers observing a high prevalence of smokers among patients affected by SCC of the nasal vestibule (which we consider distinctly to SCC of the nasal cavity and paranasal sinuses); 3) and papers observing an increased tendency towards malignant transformation of inverted papilloma in smokers. Based on this evidence, we added the following sentence to the review:

“Differently to other HNSCCs, the putative etiological role of tobacco smoking is based upon frail and dated evidence,[3] and alcohol consumption has not been demonstrated as being a risk factor for SNSCC. Several patients affected by SCC of the nasal vestibule had a history of tobacco smoking,[4,5] which probably led some authors to find an association with this risk factor when SCCs of nasal vestibule, nasal cavity, and paranasal sinuses are grouped altogether.”

“Tobacco smoking is thought to be a factor promoting transformation of Schneiderian papillomas into SNSCC.[19]

As concerns the fact that SCC could display pathological features that are found also in SNEC and SNUC we find worth specifying this characteristic of sinonasal SCC, as quite often the diagnosis of SCC is given with a certain degree of approximation also in tumors displaying overlapping features. This represent a clear limit of our current way to classify sinonasal cancers, and the reader should be made aware of the existence of these overlapping features.

  • “The information on the role of HPV in sinonasal squamous cell carcinoma is outdated. The references on this topic are old [ref 16-18]. Several high quality papers were published on the topic in 2020 and 2021 including in Head and Neck as well as Cancers. The authors should cite more recent references on this topic.”

Thank you for pointing it out. As you correctly alluded to, we recognize that recent publications have converged in suggesting an etiological role and prognostic implications with a higher level of evidence than in the publications we cited. The following sentence has been added alongside with the most relevant publications leading to this conclusion:

However, recent publications converge in stating that HPV probably has a role in determining SNSCC and implies a better prognosis compared to HPV-negative tumors.[10–12]

  • “The staging paragraph needs the AJCC 8th Ed staging for sinonasal cancers and the modifications to T staging compared to the 7th Ed. Special mention should be made with respect to T4b / non surgical disease. This is important as the authors presents later in the manuscript case examples of what would be considered non surgical disease treated with surgery (including dural resection and repair)”

Thank you for your suggestion. However, the 8th Edition of TNM classification does not include substantial changes in criteria to categorize sinonasal cancer. Moreover, T4b is no longer considered an unresectable (or non-surgical disease) by definition, and this particularly applies to sinonasal cancer, as dural infiltration designate a tumor as T4b but can be managed with surgery in the majority of cases. Based on your input, the following sentences were added to staging and surgery paragraphs, respectively:

“Another crucial step in treatment planning for sinonasal tumors is the analysis of local extension, with special reference to skull base, orbit, and infracranial spaces (i.e., pterygopalatine fossa, infratemporal fossa, and parapharyngeal space), which allows attributing a “T category” to a given sinonasal cancer according to the latest TNM classification criteria.[41]

“Of note, some tumor extensions that can be managed with surgery fall under the definition of T4b, which should thus not be automatically considered as an unresectable disease, as also stated in the latest National Comprehensive Cancer Network (NCCN) guidelines (version 3.2021).[90]

  • “The paragraph on neoadjuvant chemotherapy should include the ongoing multi institutional clinical trial EA3163 (https://ecog-acrin.org/clinical-trials/ea3163-educational-materials)”

Thank you for this valuable advice. We reported the suggested trial and other trials labelled with their ClinicalTrials.gov Identifiers.

“Several clinical trials are currently investigating the potential benefit conferred by neoadjuvant ChT in sinonasal cancers (e.g., ClinicalTrials.gov Identifiers: NCT03493425, NCT02099175, NCT02099188, NCT00707473).”

“A registered clinical trial is currently comparing definitive IMPT with IMRT for sinonasal malignancies (ClinicalTrials.gov Identifier: NCT01586767), whereas another trial is investigating the role of R2 endoscopic surgery followed by IMPT in patients affected by unresectable sinonasal cancer (ClinicalTrials.gov Identifier: NCT03274414).”

  • “Table 2 includes several categories of cases that would be considered non surgical for sinonasal squamous cell carcinomas with the corresponding surgery. This is more the author's experience than a literature review. Orbital apex, clivus, sella, falx cerebri, brain etc involvement by squamous cell carcinoma would be considered non surgical disease. Review of the current AJCC staging and NCCN guidelines would provide a more accurate view of the currently accepted surgical indications for sinonasal cancer. The NCCN guidelines are mentioned at the end of the manuscript (line 612) for follow up after treatment. There is merit in reviewing the guidelines throughout the manuscript as those are based on extensive literature review and current evidence.”

Thank you for this comment. We recognize that potential indications to surgery summarized in Table 2 are reported based on authors’ experience and literature analysis, yet many of topographical extensions are not univocally considered as resectable. Again, this aspect is a matter of debate and there is not clear evidence to define some local extensions as either resectable or non-resectable. We think that specifying that these indications are based on authors’ opinion could be adequate. Coherently, the latest NCCN guidelines state that several critical extensions “are not an absolute contraindication to resection in selected patients in whom total cancer removal is possible”. The fact that case-by-case evaluation is essential and generalization of local extension to be addressed surgically should be avoided has been explicitly reported.

“Indications to purely endoscopic, endoscopic-assisted, and non-endoscopic procedures for sinonasal cancers according to authors’ experience are summarized in Table 2. Of note, some tumor extensions that can be managed by surgery fall under the definition of T4b SNSCC, which should thus not be automatically considered as an unresectable disease, as also stated in the latest National Comprehensive Cancer Network (NCCN) guidelines (version 3.2021).[90] The most critical extensions of SNSCC, as for all sinonasal cancers, are those towards the intracranial space,[91,92] infracranial spaces,[93–97] and orbit.[49,98–100] However, advancements in surgical technique have led to narrow the spectrum of extensions to be considered as genuinely unresectable to a limited number of circumstances (e.g., invasion of both internal carotid arteries, both orbital cavities/optic nerves or brainstem). More frequently, surgery is excluded for the degree of invasiveness it would imply and the howsoever dismal prognosis. A case-by-case evaluation is thus crucial to correctly identify patients with advanced SNSCC eligible for surgery.”

  • “Reconstruction should be rewritten in a smaller paragraph and included within the surgical treatment paragraph. There is nothing unique to squamous cell carcinoma. This is rather a section purely on surgical technique. The significant relevance of the reconstruction from an oncologic perspective is that failure of reconstruction could delay adjuvant treatment which is crucial to optimize the chances of cure in moderately advanced sinonasal cancer.”

Thank you for this consideration. We recognize that reconstruction is not unique to squamous cell carcinoma. Nonetheless, the paragraphs do not explain surgical technique (which would require a much longer and off-topic discussion), rather they briefly summarize the current strategies to reconstruct post-ablative defects. In our opinion, this is not a minor issue from an oncologic standpoint, as an optimal achievement from a survival standpoint could be associated with suboptimal result from a functional/morbidity perspective if reconstruction is not properly addressed upfront. Since surgery is still the mainstay of treatment of most sinonasal carcinomas, the scientific community dealing with sinonasal squamous cell carcinomas should have an idea of what the patient has to go through as part of the surgical treatment. The aim of the review is to provide an overview of the current evidence on management of sinonasal squamous cell carcinoma: just like details on chemotherapy and radiotherapy are expected, pertinent details on surgery should be given as well. The fact that defects following ablation of nasoethmoidal/sphenoidal versus maxillary/frontal cancers bear different challenges is best expressed by dividing this discussion in 2 subparagraphs (whose word length is limited; i.e., 237 and 320, respectively).

  • “The section on adjuvant therapy is too long. The section on follow up and management of recurrent disease is too long.”

Again, yet respecting your opinion on these paragraphs, we have a different view. In our opinion, the paragraphs you referred to do not include redundant information, thus they have to be structured in order to carry the important messages and concepts through the needed length. Adjuvant therapy is an essential part of the treatment and is indicated in the large majority of patients affected by sinonasal squamous cell carcinoma. The inherent paragraph is 610-word long (i.e., 7.4% of the review); thus, we do not think it is too long. A considerable proportion of patients treated for sinonasal squamous cell carcinoma experiences a recurrence. This mandates to adequately consider how to diagnose and manage recurrent squamous cell carcinoma in a review intended to reappraise the topic. The updated inherent paragraph (which has been divided in subparagraphs to make it clearer) is 1480-word long (i.e., 17.9% of the review); thus, we do not think it is too long.
